# Genetic diversity, relatedness and inbreeding of ranched and fragmented Cape buffalo populations in southern Africa

**Deon de Jager**[1]*, **Cindy Kim Harper**[2], **Paulette Bloomer**[1]

**1** Molecular Ecology and Evolution Programme, Department of Biochemistry, Genetics and Microbiology, Faculty of Natural and Agricultural Sciences, University of Pretoria, Pretoria, Gauteng, South Africa, **2** Veterinary Genetics Laboratory, Faculty of Veterinary Science, University of Pretoria, Pretoria, Gauteng, South Africa

* dejager4@gmail.com

**Data Availability Statement:** The genotype file is available from the Dryad Digital Repository (https://doi.org/10.5061/dryad.1c2d3).

**Funding:** DdJ hereby acknowledges the financial contribution of the National Research Foundation

## Abstract

Wildlife ranching, although not considered a conventional conservation system, provides a sustainable model for wildlife utilization and could be a source of valuable genetic material. However, increased fragmentation and intensive management may threaten the evolutionary potential and conservation value of species. Disease-free Cape buffalo (*Syncerus caffer caffer*) in southern Africa exist in populations with a variety of histories and management practices. We compared the genetic diversity of buffalo in national parks to private ranches and found that, except for Addo Elephant National Park, genetic diversity was high and statistically equivalent. We found that relatedness and inbreeding levels were not substantially different between ranched populations and those in national parks, indicating that breeding practices likely did not yet influence genetic diversity of buffalo on private ranches in this study. High genetic differentiation between South African protected areas highlighted their fragmented nature. Structure analysis revealed private ranches comprised three gene pools, with origins from Addo Elephant National Park, Kruger National Park and a third, unsampled gene pool. Based on these results, we recommend the Addo population be supplemented with disease-free Graspan and Mokala buffalo (of Kruger origin). We highlight the need for more research to characterize the genetic diversity and composition of ranched wildlife species, in conjunction with wildlife ranchers and conservation authorities, in order to evaluate the implications for management and conservation of these species across different systems.

## Introduction

Wildlife ranching, where wild animals are managed in fenced areas, is practiced in various regions around the world, including North America, Europe and Africa [1–5]. Whether wildlife ranching is beneficial for conservation is often debated [6–8]. While wildlife ranching preserves habitat that would otherwise be converted for other land-use types [9], it also comes

(https://www.nrf.ac.za/), under the grant number SFH150630122321. Opinions expressed, and conclusions arrived at are those of the authors and are not necessarily to be attributed to the NRF. The funders had no role in study design, data collection and analysis, decision to publish, or preparation of the manuscript.

**Competing interests:** The authors have declared that no competing interests exist.

with the pitfalls of small population sizes and fragmentation of populations due to fencing that could result in loss of genetic diversity of many species, not just those of economic interest on the ranch [5, 10]. Furthermore, wildlife ranching often involves intensive breeding of wildlife, or intentional genetic manipulation [5], for a single, or multiple, traits. This can exacerbate the fixation or loss of alleles at a rate much higher than would occur through drift or natural selection [11]. Consequently, the frequency of alleles that are identical by descent increases in the population, thus increasing the risk of inbreeding and inbreeding depression [12]. Deleterious alleles that are in linkage disequilibrium with the genes underlying the desired traits will also be inherited by offspring and increase in frequency in the population [11]. Such intentional genetic manipulation of wildlife may decrease the evolutionary potential of species and reduce the conservation value that may be provided by privately-owned wildlife populations [5, 13].

Few studies have been done comparing genetic diversity of species on intensively managed wildlife ranches to more open or natural systems in state protected areas (reserves and transfrontier- or national parks) and extensive game ranches. A comprehensive study of populations of red deer (*Cervus elaphus*) on ranches and in a national park in Spain found no significant differences in genetic diversity [2]. In South Africa, Grobler et al. [14] found significantly lower allozyme diversity in impala (*Aepyceros melampus*) populations in smaller, managed reserves compared to a large population, although these reserves were not intensively breeding impala. In a study of blue and black wildebeest (*Connochaetes taurinus* and *C. gnou*), Grobler et al. [15] found no significant difference in genetic diversity between ranch and reference populations of these species, although this was not the aim of the study. Finally, Grobler et al. [16] used polymorphic allozyme loci and found substantially lower genetic diversity in a ranch population of Cape buffalo (*Syncerus caffer caffer*) compared to reference samples from the Kruger National Park (KNP). However, the ranch population was established with only two bulls and six cows from Addo Elephant National Park (AENP), which is now known to have relatively low genetic diversity [16, 17]. This indicates that the low diversity of the ranch population was likely due to founder effect, given that 25 years (~3.5 buffalo generations) had passed between establishment of the population and sampling.

At the time of the Grobler et al. [16] study, private ranches in South Africa could only be stocked with disease-free buffalo from AENP and zoological gardens around the world. This was due to the prevalence of bovid diseases in other potential source populations, such as KNP or Hluhluwe-iMfolozi Park (HiP), that are controlled in South Africa through strict veterinary regulations [18]. In the 1890s and early 1900s rinderpest and foot-and-mouth disease epidemics resulted in the loss of an estimated 95% of the buffalo population in southern Africa [18]. During the 1980s and 1990s, outbreaks of bovine tuberculosis and Corridor disease in the buffalo populations of South Africa (excluding AENP) prompted South African National Parks (SANParks) to start a disease-free breeding programme for KNP buffalo which was carried out between 1999 and 2007 [19]. The goal was to maintain a disease-free population outside disease-affected areas that would represent the high genetic diversity of KNP. Two of these disease-free KNP-derived populations are now maintained in Mokala National Park (MNP) and the nearby breeding centre Graspan (GNP). Consequently, the wildlife ranching industry had a new source of disease-free buffalo with presumably high genetic diversity.

Concurrently, the wildlife ranching industry in South Africa experienced rapid growth, not only due to the popularity of disease-free buffalo, but also due to the breeding, selling and eventual hunting of rare phenotypes of various antelope species, such as exotic colour variations, unique horn morphology and increased horn length [5, 20, 21]. Buffalo is one of the most expensive species and are generally bred to obtain exceptional horn length and "spread", with buffalo of East African origin being more expensive and sought after than southern African individuals due to their apparent superior phenotype in this regard [5].

In this study, we aimed to: (i) Determine how successful the SANParks disease-free breeding programme was in maintaining the high genetic diversity of KNP buffalo; (ii) Investigate whether the genetic diversity of buffalo under intensive management regimes on private ranches was significantly different compared to their source populations in national parks; (iii) Determine how the relatedness and inbreeding levels of buffalo on private ranches compared to those in national parks and; (iv) Characterize the genetic composition of private ranch buffalo populations. We discuss how the results relate to the general breeding practices on private ranches and what the implications are for the conservation and management of Cape buffalo and other wildlife species that are popular in wildlife ranching. Finally, we make recommendations regarding the genetic management of Cape buffalo in national parks in southern Africa.

## Materials and methods

### Samples and ethics statement

This study was performed in collaboration with SANParks (Project code: HARC1227) and the Ministry of Environment and Tourism, Namibia. Samples from MNP (Northern Cape, South Africa, GPS: -29.162613, 24.321083), GNP (located near MNP in the Northern Cape) and AENP (Eastern Cape, South Africa, GPS: -33.483474, 25.750269) were obtained from SAN-Parks, as blood stored in ethylenediaminetetraacetic acid (EDTA). Samples from Waterberg Plateau National Park (WPP, Namibia, GPS: -20.352369, 17.337493) and 12 private wildlife ranches (P001 –P012) were previously obtained by the Veterinary Genetics Laboratory (VGL) at the University of Pretoria. All samples were collected between 2008 and 2015. Thus, within one buffalo generation (5–7.5 years), with 99% of the samples collected between 2011 and 2015 (Table 1). Private buffalo owners send samples to the VGL for individual genotyping and parentage analysis. The number of samples from each locality is shown in Table 1. Permission

**Table 1. Sample information and selected summary statistics of Cape buffalo from the 16 localities included in this study.**

| Locality | Collection year | $N_C$* | $N$ | $N_A$ | $P_A$ |
|---|---|---|---|---|---|
| AENP | 2008, 2013, 2014 | 800 | 79 | 37 | 0 |
| GNP | 2011, 2012, 2013 | 80 | 21 | 68 | 1 |
| MNP | 2011, 2012, 2013, 2014 | 400 | 35 | 80 | 1 |
| WPP | 2014 | 600 | 95 | 62 | 0 |
| P001 | 2012, 2013, 2014 | N/A | 153 | 89 | 2 |
| P002 | 2012, 2014 | N/A | 308 | 93 | 1 |
| P003 | 2014 | N/A | 21 | 59 | 0 |
| P004 | 2009, 2011, 2012, 2013, 2014 | N/A | 262 | 99 | 1 |
| P005 | 2013, 2014 | N/A | 57 | 86 | 0 |
| P006 | 2013 | N/A | 164 | 96 | 2 |
| P007 | 2014 | N/A | 17 | 64 | 0 |
| P008 | 2014 | N/A | 54 | 78 | 1 |
| P009 | 2011, 2013, 2014 | N/A | 99 | 85 | 1 |
| P010 | 2013 | N/A | 35 | 70 | 1 |
| P011 | 2015 | N/A | 22 | 69 | 0 |
| P012 | 2014 | N/A | 37 | 77 | 0 |

*Approximate figures. $N_C$: Population census size. $N$: Sample size. $N_A$: Total number of alleles across the 11 microsatellite markers used in this study. $P_A$: Private (or unique) alleles present in that population. N/A: Not available.

was obtained from the Namibian Department of Environmental Affairs and Tourism and each private ranch owner to use the genotype data of their samples in this study. Data from the private ranches, such as sample and ranch names, have been anonymized for this study. Ranch locations have also been omitted as this may compromise the anonymity of the participating ranches. The location of the ranches was not important for the analyses and conclusions of the study, except to say that all ranches are located in South Africa. This project was approved by the Animal Ethics Committee of the University of Pretoria (code: ec005-16).

## DNA extraction and microsatellite genotyping

DNA was extracted from EDTA-blood samples from AENP, MNP and GNP using the PrepFiler® Forensic DNA Extraction Kit (Applied Biosystems, CA, USA) and genotyped at 18 variable microsatellite loci and a sex marker using the following PCR conditions: initial denaturation at 95˚C for 3 min, 35 cycles of 95˚C for 15 s, 60˚C for 30 s and 72˚C for 30 s, followed by a final extension step of 72˚C for 10 min. Fragments were separated on an ABI 3500xl Genetic Analyzer (Applied Biosystems) together with GeneScan™ LIZ™ 500 dye Size Standard (Applied Biosystems). Allele binning and scoring was performed using STRand version 2.4.110 [22] (http://www.vgl.ucdavis.edu/informatics/strand.php). Samples from WPP and the 12 private ranches were previously genotyped at between 15 and 18 loci. Microsatellite loci that were not common to all data sets were removed, thus the final data set consisted of 1,459 buffalo genotyped at 11 microsatellite loci and a sex marker (S1 Table). The microsatellite loci used in this study have previously been deemed satisfactory in terms of conforming to Hardy-Weinberg equilibrium in this exact data set [23].

## Study system and population histories

All sampling localities in this study are geographically isolated. The AENP population is a remnant population of buffalo that was fenced in during 1931 when the borders of AENP were set up and remains free of bovid diseases [17]. The number of buffalo fenced in when the original borders were set up is not known. However, 130 buffalo were removed in 1981, reducing the population to 75 individuals. By 1983 the population had increased to approximately 220 individuals, but was reduced again to 52 buffalo in 1985, potentially due to drought during that time (*Pers. Comm.* D. Zimmerman 2015). The census size reported in 1998 was 85 buffalo [17]. The current census size is approximately 800 buffalo (Table 1). There are no records of any human-mediated introduction of buffalo to AENP at any point in the history of this population (*Pers. Comm.* D. Zimmerman 2015).

The GNP and MNP populations were established through the SANParks disease-free breeding programme in 1999. Thereafter, each was supplemented with yearlings, after disease testing, from the KNP breeding group until 2007 (*Pers. Comm.* D. Zimmerman 2015). The breeding group consisted of buffalo (approximately 140 cows and 10 bulls) that originated mainly from northern KNP and contained in a fenced-off camp in KNP (*Pers. Comm.* D. Zimmerman 2015). No introductions have been made to GNP or MNP since 2007 (*Pers. Comm.* D. Zimmerman 2015). The current census sizes of GNP and MNP are approximately 80 and 400 buffalo, respectively (Table 1).

A disease-free buffalo population unrelated to those managed by SANParks was established in Namibia in 1981 in what is now known as Waterberg Plateau National Park (WPP). The buffalo population in WPP was founded with seven buffalo from AENP in 1981, with additional introductions of 26 AENP buffalo between 1986 and 1991 (*Pers. Comm.* M. Lindeque 2017). Two introductions of five and six buffalo from the Willem Pretorius Nature Reserve (WPNR) in the Free State Province of South Africa occurred in 1985 and 1986, respectively

(*Pers. Comm.* M. Lindeque 2017). Although a detailed history of WPNR buffalo population could not be established, it has been reported that this population was founded with two bulls and six cows originating from AENP around 1986 [16]. Lastly, four buffalo were introduced in 1986 from a Namibian game dealer who had imported buffalo from a zoo in the Czech Republic. The origin of the buffalo from the zoo are suspected to be East African (most likely Tanzania or Kenya), but this could not be confirmed. There have been no additional introductions of buffalo to WPP since 1991 (*Pers. Comm.* M. Lindeque 2017). The current census size of WPP is approximately 600 buffalo (Table 1).

The individual histories of the private ranch buffalo populations included in this study are not known. However, many of the buffalo in the wildlife ranching industry were sourced from the AENP population (disease-free), as well as from the disease-free KNP breeding programme (GNP and MNP). The introduction of buffalo from other areas, such as East Africa and Zimbabwe, onto private ranches is thought to occur. Therefore, the private ranch buffalo populations in this study represent various combinations of disease-free buffalo from AENP, GNP, MNP and potentially or parts of southern Africa and/or East Africa.

## Population statistics

The number of alleles ($N_A$), allelic richness ($A_R$) and inbreeding coefficient ($F_{IS}$) of each sampling location were calculated in R v3.1.3 [24], using the divBasic function in the diveRsity package v1.9.90 [25]. The divBasic function calculates $A_R$ by normalising all populations to the smallest sample size and subsampling 1,000 times, with replacement, and thus provides 95% confidence intervals. This is considered one of the most precise ways to obtain unbiased estimates of $A_R$ when sample sizes are unequal [26]. The number of private alleles ($P_A$) in each population was determined from the tables of allele frequencies calculated in Genetix v4.05.2 [27]. Observed ($H_O$) and expected heterozygosity ($H_E$) were calculated in Cervus v3.0.7 [28]. Effective population size ($N_e$) for each sampling locality was estimated using NeEstimator v2.01 [29]. The linkage disequilibrium method was used at a critical value of 0+ and 95% confidence intervals were calculated by jack-knifing over loci 1,000 times. $N_e$ was also estimated for GNP-MNP combined and all private ranches combined.

## Relatedness and individual inbreeding

Pairwise relatedness was estimated in COANCESTRY v1.0.1.8 [30] using the likelihood estimator, TrioML [31]. This estimator was chosen as it had the smallest variance of the seven estimators available in COANCESTRY (S2 Table) and the estimates produced are between zero and one, thereby facilitating interpretation. Relatedness was estimated independently for each sampling locality, using the allele frequencies of that locality, and without accounting for inbreeding, as most localities showed little or no evidence of inbreeding. For example, while AENP had low genetic diversity, its population-level $F_{IS}$ was not significantly greater than zero, and while P006 had an $F_{IS}$ significantly greater than zero, it had high genetic diversity. Two statistics to describe distributions, the skewness (a measure of asymmetry) and kurtosis (a measure of how strongly the distribution is tailed), were calculated for the distribution of pairwise relatedness values, to assist in more robust interpretation. These were calculated using the package moments v0.14 (https://cran.r-project.org/web/packages/moments/index.html) in R v3.5.0 [32]. In order to determine whether there was a disproportionate number of close relatives (second- and first-order relatives) in private ranches compared to national parks, possibly due to breeding practices, the proportion of pairwise relatedness values greater than or equal ($\geq$) to 0.25 was calculated for each sampling locality. Mean relatedness within sexes per locality was also calculated to determine whether there was any sex-bias in relatedness distributions.

The genotype at the amelogenin sex marker was used to group individuals into sexes. Individual inbreeding coefficients, $F$, were estimated independently for each sampling locality in COANCESTRY using the TrioML method, by selecting the "Account for Inbreeding" option.

## Population differentiation

To investigate genetic differentiation between sampling localities, the population differentiation estimator, Jost's $D$ ($D_{JOST}$) [33], and Weir and Cockerham's fixation index ($F_{ST}$) [34], were calculated, for all pairs of localities, in R v3.1.3 using the diffCalc function in the diveRsity package. The 95% confidence intervals for these statistics were calculated with the same function, using 999 bootstrap replicates, to determine significant deviation from zero. Correlation between $D_{JOST}$ and $F_{ST}$ was calculated in R v3.5.0 with the cor.test function using the non-parametric Spearman's rank correlation, *rho*, since the data were not normally distributed (as shown by a Shapiro-Wilk normality test in R). Genetic structure was also investigated using STRUCTURE v2.3.4 [35] and a discriminant analysis of principal components (DAPC) [36] using the package adegenet v2.1.1 [37] in R v3.5.0. The STRUCTURE analysis was conducted on the full data set, as well as a data set where relatives were removed. See S1 Appendix for details of these analyses.

The R scripts used to create most of the figures in this manuscript and to perform the DAPC analysis are available at the following URL: https://github.com/DeondeJager/Rscripts-for-buffalo-microsat-paper. Rstudio v1.2.5033 [38] was used for all R-based analyses.

## Results

### Genetic diversity of national parks and private ranches

In this study, Graspan (GNP) and Mokala National Park (MNP) were used as "benchmark" localities in terms of genetic diversity due to the Kruger National Park (KNP) origin of these buffalo. Addo Elephant National Park (AENP) had the fewest number of alleles, despite having the seventh largest sample size in the study. Nine of 11 private alleles were present in the private ranch samples, with the remaining two private alleles present in GNP and MNP (Table 1).

The allelic richness ($A_R$) analysis showed that AENP had significantly lower genetic diversity than any other population in this study (Fig 1A, S3 Table). WPP and P003 had significantly lower $A_R$ than the mean across all the populations, whereas P001, P002 and P006 had significantly higher $A_R$ than the mean (Fig 1A). When the mean was calculated without AENP, only P001 remained with significantly higher $A_R$. Despite its low $A_R$, AENP showed no significant population-level inbreeding ($F_{IS}$, Fig 1B). Conversely, P006 showed a low, but significant $F_{IS}$ value, with an excess of homozygotes, despite having the highest $A_R$. P007 showed an excess of heterozygotes, with an $F_{IS}$ significantly less than zero. The observed and expected heterozygosity (Fig 1C) values had large standard deviations, but mirrored the inbreeding coefficients, where a lower observed than expected heterozygosity corresponded to a positive inbreeding coefficient and vice versa.

Effective population size ($N_e$) estimates were similar for most localities (range: 3.3–48.8), as well as for GNP-MNP and all private ranches combined (88.7 and 81.9, respectively) (S1 Fig, S3 Table). AENP had an $N_e$ estimate of 24.2 (S3 Table), although this was not significantly lower than GNP-MNP, based on (slightly) overlapping 95% confidence intervals (S1 Fig).

### Relatedness and individual inbreeding

The pairwise relatedness ($r$) values had an L-shaped distribution, with a slight increase in frequency of values around $r = 0.5$ in all localities, albeit to varying degrees (Fig 2). Mean pairwise

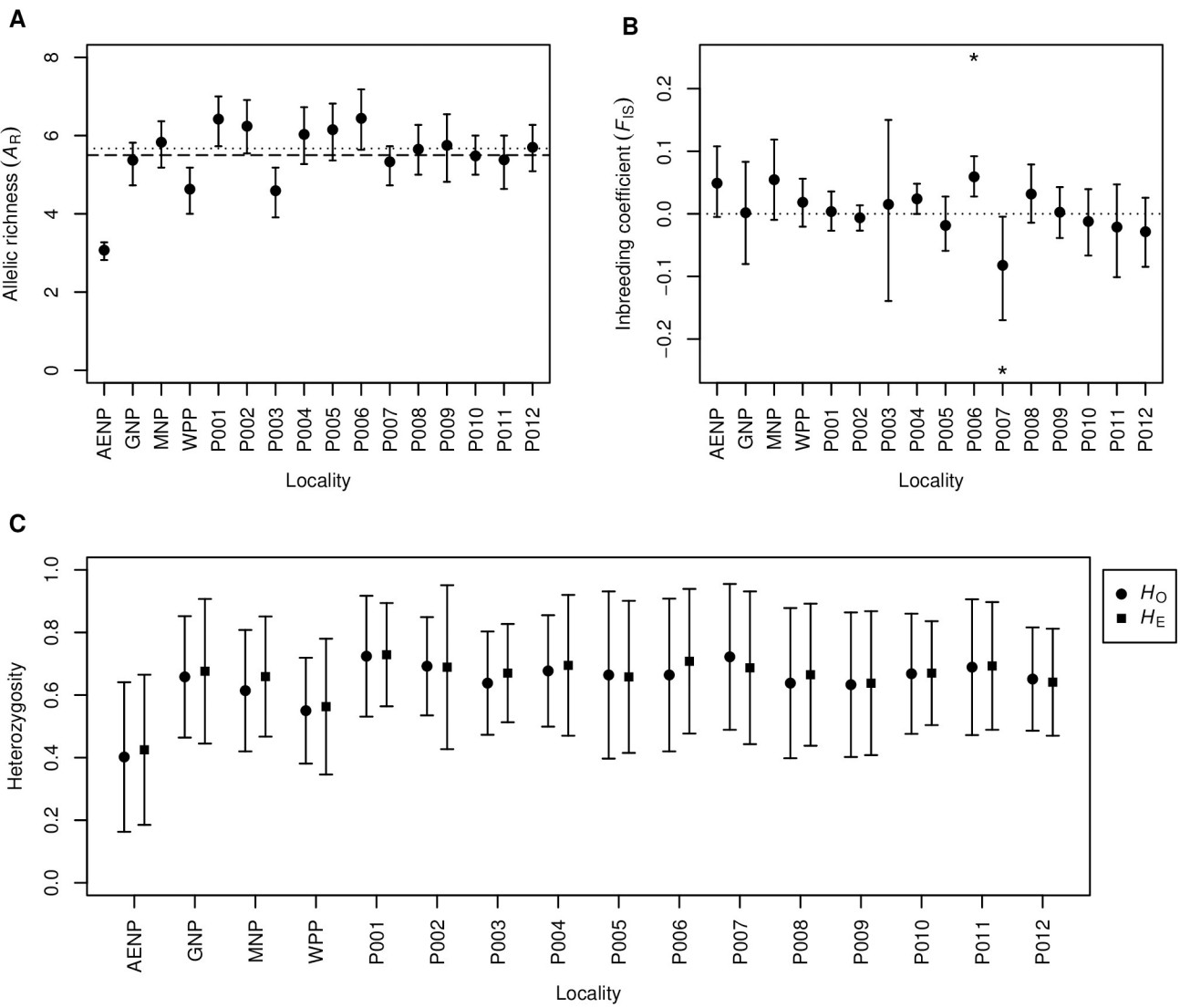

**Fig 1. Genetic diversity indices of the 16 sampling localities analysed in the present study. A** Allelic richness, where the mean is shown by the dashed line and the mean without AENP is shown by the dotted line. **B** Inbreeding coefficient, where an asterisk (*) indicates significant deviation from zero (dotted line). **C** Observed ($H_O$) and expected ($H_E$) heterozygosity. Vertical bars indicate 95% confidence intervals on plots A and B, and standard deviation on plot C.

relatedness was generally low, where P003 (0.110) and AENP (0.096) had the highest and GNP (0.044) and MNP (0.049) had the lowest mean relatedness (Fig 2, S4 Table). The positive, and relatively high, skewness values confirmed the distributions were right-tailed and highly asymmetrical (range: 1.75–3.31, S4 Table). A normal distribution has a skewness of zero and kurtosis of three. The kurtosis values of GNP and MNP were the highest at 14.64 and 12.56 (S4 Table), meaning their relatedness distributions had the most extreme tails, i.e. their tails were the "thinnest". As stated by Bradley [39]: "The coefficients of skewness and kurtosis of an L-shaped distribution increase rapidly as the long positive tail of the distribution becomes thinner and thinner. . .". In other words, GNP and MNP had the lowest proportion of close relatives, while those localities with lower kurtosis values, such as P003 (3.49) and AENP (5.14), had a larger proportion of close relatives.

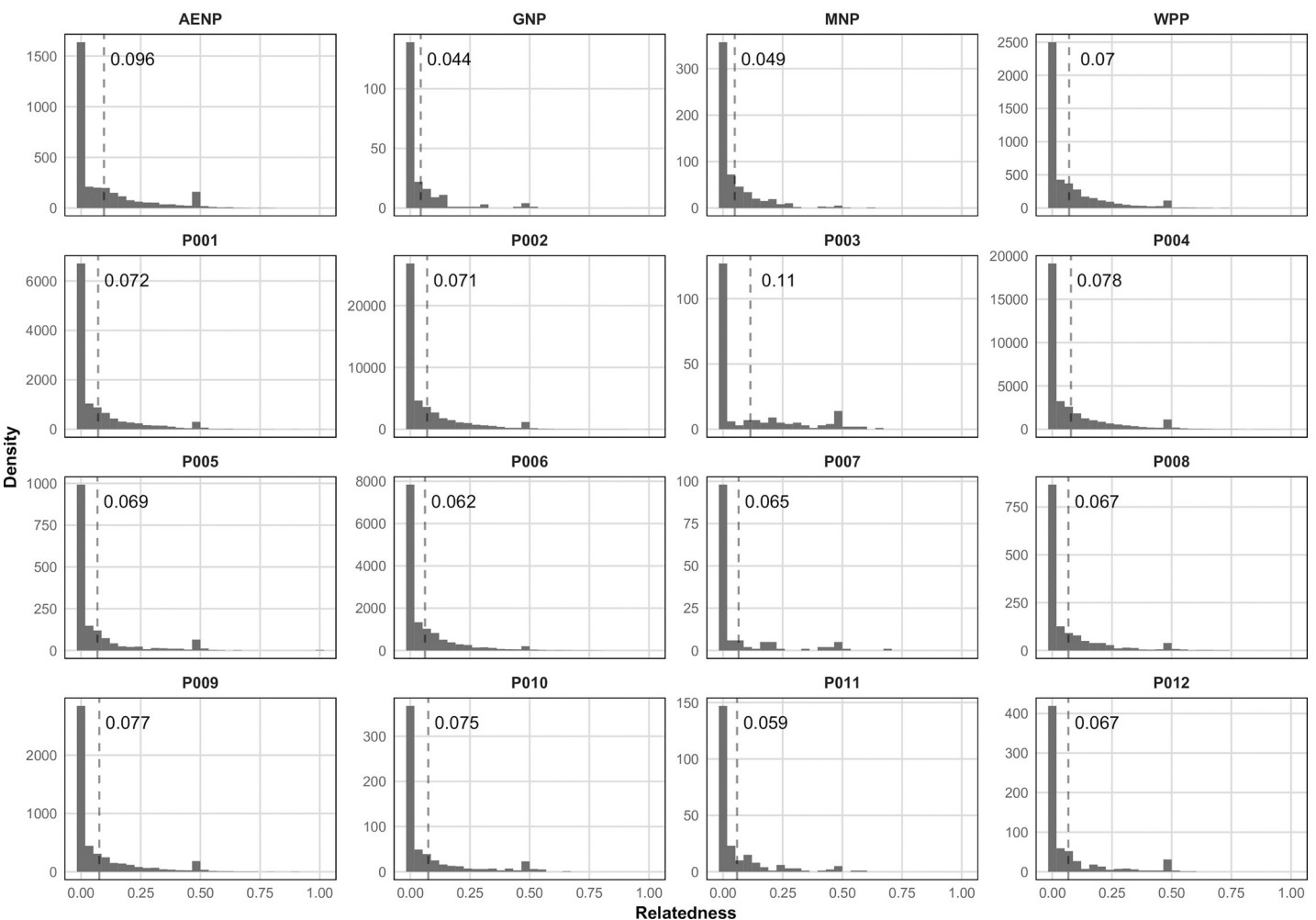

**Fig 2. Pairwise relatedness distributions per locality.** Density histograms were constructed with ggplot2 v3.3.0 [40] in R, with 30 bins for all histograms. The dashed line on each plot indicates the mean relatedness, with the text inset showing the value of the mean. Note that the y-axes scales are independent between plots to aid in visualization of the distribution for each locality, due to the large range of sample sizes in this data set.

Indeed, this was illustrated by the strong negative correlation between kurtosis of the relatedness distributions and the proportion of pairwise relatedness values $\geq 0.25$ (Fig 3). The proportion of close relatives, as estimated by the proportion of $r$ values $\geq 0.25$, was not significantly higher in private ranches (mean = 10.0%) compared to national parks (mean = 8.1%) ($p$-value = 0.2, one-tailed t-test). However, private ranches generally had a proportion of close relatives approximately twice the value seen in GNP (4.8%) and MNP (4.2%), while this value in AENP (14.5%) was approximately three times higher than the latter two (S4 Table). In general, males were more related to each other than females, except for P003 and P011, although this difference was not significant when the mean was calculated across all localities ($p$-value = 0.077, one-tailed t-test) (S5 Table).

Individual inbreeding ($F$) estimates were substantially higher in AENP (mean $F$ = 0.156) compared to the other localities, except for P006 which had a mean $F$ of 0.112 (Fig 4, S4 Table). The higher levels of $F$ in these two localities was also evident in their distributions in Fig 4, with larger interquartile ranges and higher 75% quantiles than all other localities. The remainder of the localities had relatively low inbreeding estimates, with means ranging between 0.023 and 0.097 (S4 Table).

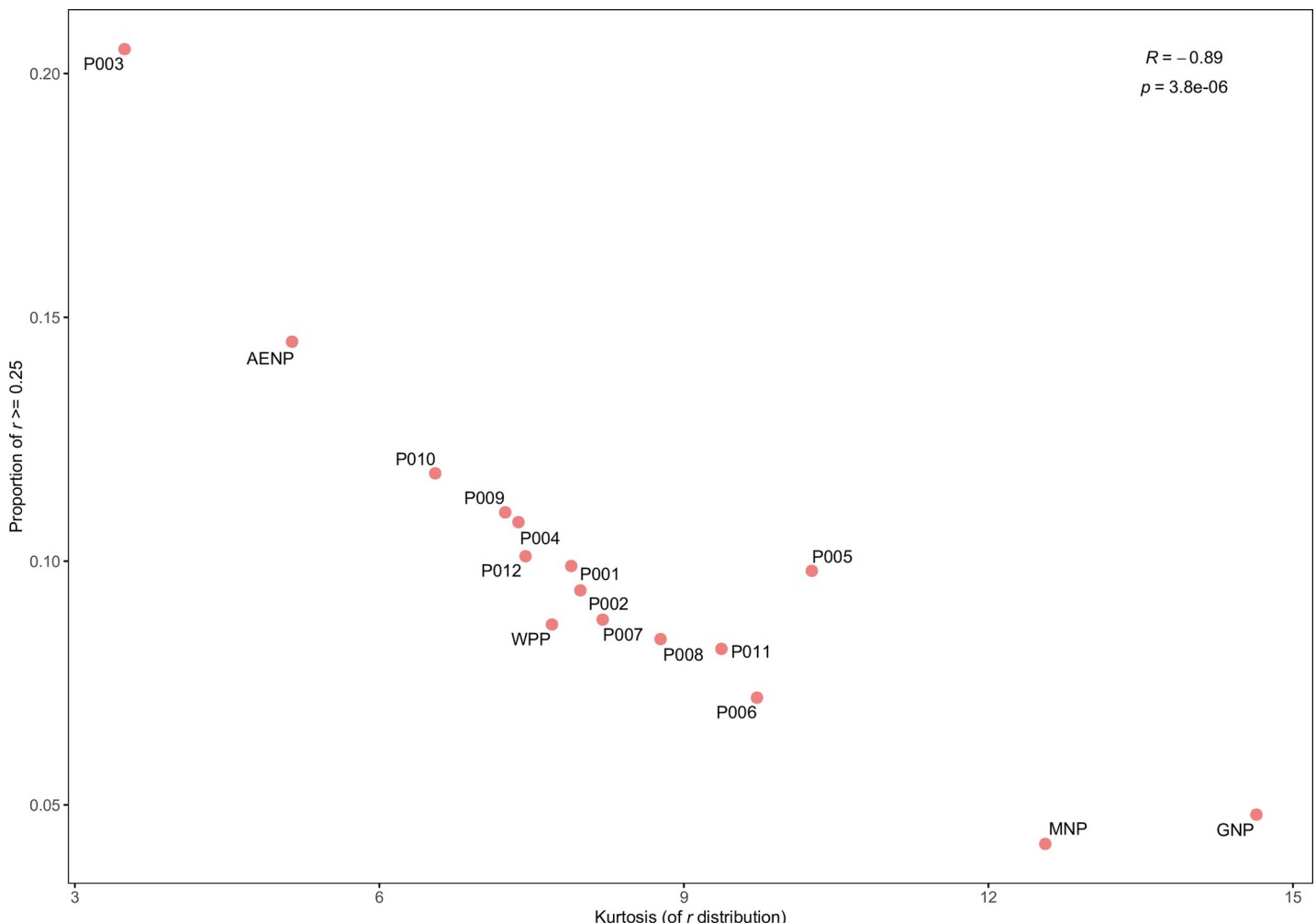

**Fig 3. Scatterplot showing the correlation between kurtosis and proportion of relatedness values $\geq$ 0.25.** Pearson's correlation coefficient ($R$) and the associated $p$-value are shown.

## Population differentiation

The majority of the 120 population pairs were significantly differentiated based on the $D_{\text{JOST}}$ and $F_{\text{ST}}$ analyses (Table 2). These two metrics were significantly correlated, Spearman's $rho$ = 0.90, $p$-value $< 2.2\text{e}^{-16}$. Differentiation estimates were deemed significant if the 95% confidence interval did not encompass zero (S6 Table). Only four pairs were not significantly differentiated based on $F_{\text{ST}}$, namely GNP-MNP, P003-P006, P003-P008 and P003-P011, while ten pairs were not significantly differentiated based on $D_{\text{JOST}}$ (Table 2). In terms of the national parks, GNP and MNP were more similar to each other than to any other population and their differentiation ($D_{\text{JOST}}$) was not significant, while AENP and WPP were also more similar to each other than to any other populations, but were still significantly differentiated. Both results were unsurprising given the shared origin of GNP and MNP buffalo and that AENP provided founders for WPP. The mean $F_{\text{ST}}$ across all populations was 0.077 and the mean $D_{\text{JOST}}$ was 0.116. Between the four national parks, the mean $F_{\text{ST}}$ was 0.147 and mean $D_{\text{JOST}}$ was 0.184. The corresponding values among the private buffalo populations only were 0.055 and 0.098. AENP was the most differentiated from the private ranches, with a $D_{\text{JOST}}$ of

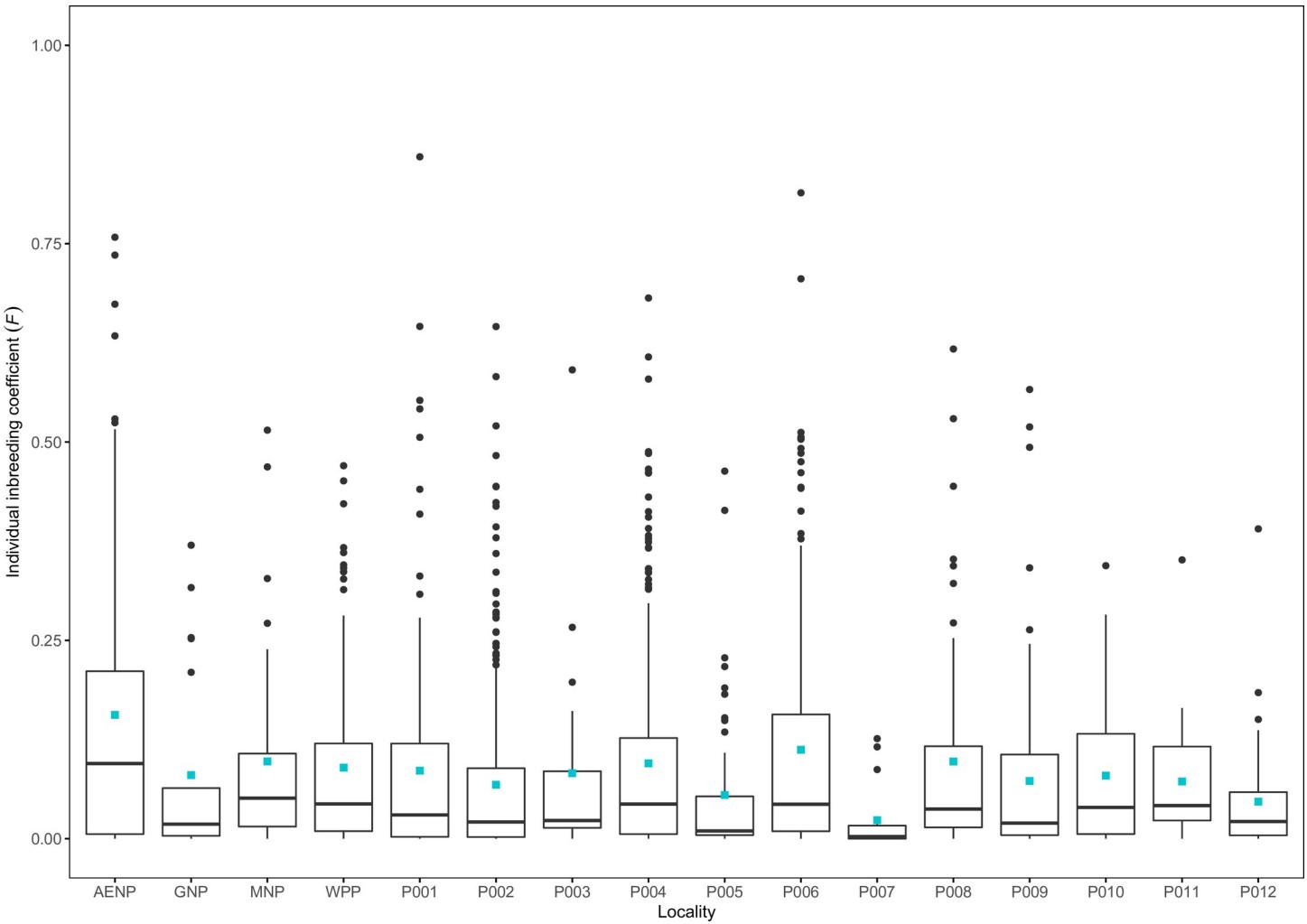

**Fig 4. Box and whisker plots showing the distribution of individual inbreeding (*F*) estimates.** The lower and upper hinges of the box indicate the 25% and 75% quantiles, respectively, with the horizontal line in the box showing the median. The lower whisker shows the smallest observation greater than or equal to the lower hinge minus 1.5 times the interquartile range and the upper whisker shows the largest observation less than or equal to the upper hinge plus 1.5 times the interquartile range. Points indicate outliers beyond the whiskers. The turquoise squares show the mean.

0.203 (mean calculated across P001 –P012), followed by WPP (0.161), MNP (0.089) and finally GNP (0.080).

The STRUCTURE analysis of the full data set revealed $K = 2$ as the most likely (based on Delta$K$), while the analysis with the relatives removed data set revealed $K = 3$ as the most likely number of genetic clusters (based on both the log likelihood of the data and Delta$K$) (S2 Fig). STRUCTURE assumes Hardy-Weinberg equilibrium (HWE), and linkage equilibrium between loci, within populations [35]. In the full data set, five private ranches did not conform to HWE, after Bonferroni corrections (S7 Table). However, in the relatives removed data set, all localities conformed to HWE (S7 Table). In the full data set, some pairs of loci were not in linkage equilibrium after Bonferroni corrections were performed, although there was no pair consistently in linkage disequilibrium (S8 Table). However, in the relatives removed data set, all pairs of loci were in linkage equilibrium in all localities (S8 Table). The relatives removed data set therefore satisfied the assumptions of STRUCTURE and thus we concluded that $K = 3$ represents the most likely number of genetic clusters in this data set.

**Table 2. Pairwise $D_{JOST}$ (above diagonal) and $F_{ST}$ (below diagonal) values for the 16 sampling localities in this study.**

| - | AENP | GNP | MNP | WPP | P001 | P002 | P003 | P004 | P005 | P006 | P007 | P008 | P009 | P010 | P011 | P012 |
|---|---|---|---|---|---|---|---|---|---|---|---|---|---|---|---|---|
| AENP | - | 0.289 | 0.303 | 0.038 | 0.222 | 0.166 | **0.066** | 0.131 | 0.287 | 0.137 | 0.221 | 0.095 | 0.307 | 0.310 | 0.207 | 0.286 |
| GNP | 0.255 | - | **0.003** | 0.227 | 0.112 | 0.136 | **0.077** | 0.143 | 0.058 | 0.064 | 0.113 | 0.090 | **0.019** | 0.045 | 0.052 | 0.048 |
| MNP | 0.257 | **0.005** | - | 0.246 | 0.141 | 0.173 | 0.125 | 0.147 | 0.055 | 0.081 | 0.086 | 0.111 | 0.025 | **0.028** | 0.079 | **0.020** |
| WPP | 0.053 | 0.156 | 0.155 | - | 0.138 | 0.110 | **0.049** | 0.124 | 0.233 | 0.093 | 0.192 | 0.081 | 0.274 | 0.244 | 0.141 | 0.255 |
| P001 | 0.144 | 0.060 | 0.074 | 0.079 | - | 0.029 | 0.081 | 0.054 | 0.146 | 0.068 | 0.055 | 0.086 | 0.167 | 0.138 | 0.047 | 0.164 |
| P002 | 0.135 | 0.072 | 0.082 | 0.074 | 0.016 | - | 0.061 | 0.057 | 0.148 | 0.057 | 0.071 | 0.066 | 0.175 | 0.154 | 0.074 | 0.178 |
| P003 | 0.129 | 0.069 | 0.075 | 0.069 | 0.054 | 0.059 | - | **0.076** | 0.124 | **0.012** | 0.129 | **0.009** | 0.130 | 0.111 | 0.079 | 0.167 |
| P004 | 0.112 | 0.070 | 0.078 | 0.075 | 0.025 | 0.029 | 0.050 | - | 0.134 | 0.043 | 0.058 | 0.040 | 0.154 | 0.173 | 0.085 | 0.139 |
| P005 | 0.248 | 0.038 | 0.038 | 0.159 | 0.064 | 0.067 | 0.085 | 0.064 | - | 0.124 | 0.118 | 0.120 | 0.067 | 0.074 | 0.088 | 0.104 |
| P006 | 0.106 | 0.041 | 0.049 | 0.055 | 0.028 | 0.034 | **0.014** | 0.025 | 0.056 | - | 0.072 | 0.023 | 0.119 | 0.138 | 0.044 | 0.137 |
| P007 | 0.229 | 0.076 | 0.080 | 0.131 | 0.028 | 0.041 | 0.090 | 0.047 | 0.075 | 0.045 | - | 0.111 | 0.119 | 0.184 | 0.098 | 0.137 |
| P008 | 0.099 | 0.059 | 0.062 | 0.052 | 0.038 | 0.035 | **0.022** | 0.025 | 0.065 | 0.014 | 0.069 | - | 0.13 | 0.12 | 0.07 | 0.16 |
| P009 | 0.247 | 0.025 | 0.021 | 0.169 | 0.078 | 0.083 | 0.094 | 0.076 | 0.047 | 0.061 | 0.076 | 0.073 | - | 0.059 | 0.095 | 0.032 |
| P010 | 0.272 | 0.026 | 0.023 | 0.165 | 0.069 | 0.077 | 0.090 | 0.075 | 0.046 | 0.061 | 0.092 | 0.068 | 0.031 | - | 0.086 | 0.034 |
| P011 | 0.198 | 0.029 | 0.037 | 0.111 | 0.030 | 0.043 | **0.042** | 0.042 | 0.050 | 0.024 | 0.062 | 0.041 | 0.060 | 0.04 | - | 0.07 |
| P012 | 0.265 | 0.037 | 0.018 | 0.173 | 0.082 | 0.090 | 0.104 | 0.071 | 0.057 | 0.066 | 0.094 | 0.077 | 0.032 | 0.03 | 0.04 | - |

Bold, underlined values indicate population pairs that were not significantly differentiated for the relevant statistic, based on the 95% confidence intervals (S6 Table). All other population pairs were significantly differentiated.

The individual assignment plots at both $K = 2$ and $K = 3$ showed that the private ranches predominantly constituted buffalo originating from AENP and GNP-MNP, as expected given the known history of the private ranch populations (S3 Fig). However, at $K = 3$, some localities showed a signal from a third, unsampled gene pool. P001 and P002, in particular, showed a substantial contribution from this third gene pool, while P004 and P011 also had a fairly strong contribution. In the national parks, WPP showed a not insignificant signal from this third gene pool, but clustered predominantly with AENP, as expected, while AENP was mostly homogeneous. GNP and MNP clustered together and showed some variation contributed from AENP and the third gene pool (S3 Fig).

The discriminant analysis of principal components (DAPC) at $K = 3$ (the optimum $K$ as determined above) showed distinct clustering of samples corresponding to those identified in the STRUCTURE analysis, namely an AENP cluster, GNP-MNP cluster and a third, unidentified gene pool "Other" (S4 Fig).

## Discussion

In this study, we evaluated and compared the genetic diversity, relatedness and inbreeding, as well as characterized the structure of disease-free Cape buffalo (*Syncerus caffer caffer*) in private ranches and national parks in southern Africa. Disease-free buffalo populations in southern Africa could be important reservoirs of genetic diversity for the species if future disease outbreaks occur on the scale of those of the 1890s/1900s [19]. The unique population histories of each sampling locality offered an opportunity to investigate how recent (GNP and MNP) and older (AENP) population contractions, as well as admixture of different gene pools (WPP and private ranches) affected the genetic diversity of the populations. Furthermore, given the more intensive management of buffalo populations on private ranches, in general, as compared to national parks, it was important to determine whether these practices may have

affected the genetic diversity, and how the relatedness and inbreeding distributions of these populations compare against more natural or less managed populations.

The high allelic richness, effective population size ($N_e$), low inbreeding coefficient ($F_{IS}$) and low relatedness in GNP and MNP indicated that the disease-free breeding programme appeared to have been designed in a robust enough manner to prevent a significant founder effect. The observed heterozygosity of GNP and MNP (0.66 and 0.61, respectively) was slightly lower than that previously estimated in the north of KNP (0.71–0.75), but overlapped with the distribution of heterozygosity throughout the entire park (0.62–0.75) [17, 41–44]. Thus, short of a direct comparative study between GNP-MNP and KNP using the same loci, we can conclude that the disease-free breeding programme has likely maintained the high genetic diversity of KNP buffalo in the GNP and MNP populations.

The older population contraction (due to the disease outbreaks in the 1890s/1900s) in AENP was reflected in the significantly lower allelic richness ($A_R$), the low heterozygosity, low $N_e$ and high individual inbreeding ($F$) estimates observed in this population. The observed heterozygosity in AENP (0.40) was slightly lower than previously reported (0.48) by O'Ryan et al. [17], who used seven loci compared to 11 used here. There was no overlap of loci between the studies. While the observed reduction in heterozygosity of AENP may be due to the different loci used, it is likely that the genetic diversity of this population decreased in the last two decades, given the severity of the known population contraction that occurred and the fact that there has been no gene flow into this population between O'Ryan et al. [17] and the current study (*Pers. Comm*. D. Zimmerman 2015).

The admixed populations in this study (WPP and the private ranches) had genetic diversity statistically equivalent to that of GNP and MNP, although WPP and P003 had significantly lower $A_R$ than the mean across all populations. However, the significantly higher $A_R$ in WPP, compared to AENP, the population from which the majority of its founders originated, was most likely a consequence of unique variation contributed by the non-AENP/Czech Republic zoo/East African buffalo. This highlighted the positive effect that even a few breeding migrants (whether natural or human-mediated) can have on the genetic diversity of a population, however the risk of outbreeding depression should also be taken into account when mixing individuals from isolated populations [49].

Inbreeding coefficients were low for all populations, with only P006 and P007 significantly deviating from zero, in opposite directions. P006 showed an excess of homozygosity (low, but significantly positive $F_{IS}$) and P007 showed an excess of heterozygosity (significantly negative $F_{IS}$). The latter result for P007 was most likely a result of the small sample size from this population ($N = 17$) making statistical inferences less robust, as indicated by the large confidence intervals around the $F_{IS}$ estimate. P006 was an anomaly, because it had high genetic diversity, low mean relatedness, high kurtosis and a low proportion of relatedness values $\geq 0.25$, but still had a significantly positive $F_{IS}$ value and a relatively high mean individual inbreeding value ($F = 0.112$). One potential explanation for these observations follows. On private ranches with intensive management, stud breeding is often practiced, where a single male is generally the only breeding bull in a breeding group. This breeding strategy may have been implemented for several consecutive generations in the P006 population without sufficient turnover of cows in the breeding group, thus resulting in significant inbreeding. The high genetic diversity and low relatedness could be a result of recent introductions of newly acquired buffalo that were sampled and included in this study before they could contribute their genetic variation to the population, and thus before they could have reduced the inbreeding levels in P006. The fact that this $F_{IS}$ deviation and high individual $F$ values were not seen in any of the other private ranches could be due to a faster turnover of breeding bulls and/or cows, or that the samples included in this study were more randomly sampled from those populations.

Given the high genetic diversity of the private ranches, the relatively low estimates of $N_e$ were somewhat surprising. The private ranches had $N_e$ estimates close to those estimated for AENP and WPP. The $N_e$ in AENP was only 3% of the population census size, while the $N_e$ of WPP was only 4.2% of the census size. Thus, while the Czech Republic zoo buffalo may have increased the genetic diversity of WPP, the effect of the population decline that its main source population (AENP) experienced had not been completely negated. For comparison, the $N_e$ of GNP-MNP was 18% of their combined population census size, which is comparable to the $N_e$/ $N_c$ ratios reported for KNP by van Hooft et al. [44] and O'Ryan et al. [17] of 26% and 10–30%, respectively. The low $N_e$ estimated for the private ranches may reflect the management approaches on these ranches. However, the $N_e$ estimates in this study should be interpreted with caution, particularly for the private ranches, since these populations are atypical and do not necessarily satisfy the assumptions of typical population genetics models.

The L-shaped relatedness distributions indicated that most buffalo from each sampling locality were unrelated. This was a somewhat surprising result given the observed social structure that exists in unmanaged buffalo herds, where females mate with one or a few (in large herds) dominant bulls [45, 46], and the genetic evidence from herds in KNP that supports this hypothesis [44]. Therefore, the expectation was that there would be a relatively high proportion of close relatives in each locality, and particularly in private ranches, due to the breeding strategies employed. However, the high kurtosis values and low proportions of close relatives showed that this was not the case. There was a slight increase in the frequency of first-order relatives ($r$ = ~0.5) indicating some family structure was present, albeit to a lower extent than expected.

These observations may be explained by several individual factors, or a combination thereof. First, the expected ecological and genetic patterns of buffalo herds may not hold true in the case of private ranches, and even in some smaller natural populations, due to the breakdown of natural demographic processes and social structure. Second, the samples included in this study may have originated from multiple herds within each locality (unfortunately, sampling information at this scale was not available). This would explain why most buffalo within localities were unrelated, as well as the lower-than-expected proportion of close relatives in each locality. The increase in frequency of relatedness values around $r$ = 0.5 could thus be even more pronounced in individual herds than was observed here. Lastly, the relatively frequent introduction of buffalo on private ranches (immigrants) and removal of other buffalo (emigrants) would result in lower overall relatedness and lower proportions of close relatives than expected in these populations. Only a small number of breeding migrants per generation are necessary to affect the genetic diversity and relatedness of a population [17]. This may also explain the unexpected relatedness results in GNP-MNP, which showed some evidence of (indirect) gene flow from an unsampled population- likely via their source population, KNP (see discussion of STRUCTURE results below).

Females were expected to be more related to each other than males, given their lower interherd migration levels (5–20% per generation in females vs 100% in males) [44]. Again, the results indicated that this was not the case for the localities in this study, where relatedness was generally higher within males rather than females, although overall this difference was not significant. Perhaps the non-conformity of private ranches to this sex-biased relatedness hypothesis is not entirely surprising, given that the ranches do not represent a natural system in the way that AENP does, for example. Thus, not seeing this sex-biased relatedness pattern in a more natural population such as AENP was more surprising but could again be explained by considering that samples may have originated from multiple herds.

Interestingly, samples from P003 may, in fact, represent a single herd, as the relatedness parameters of this locality most closely matched the expectations discussed above. P003 had

the highest mean relatedness (0.115), the lowest skewness and kurtosis (thus the most even relatedness distribution), the highest proportion of close relatives (20.5%) and a substantially higher mean relatedness between females (0.158) than between males (0.095), while still maintaining low individual inbreeding levels (mean = 0.083).

While $D_{JOST}$ is generally a more accurate measure of genetic differentiation than $F_{ST}$ [33], we predominantly compare $F_{ST}$ here, as most previous buffalo studies did not compute $D_{JOST}$, except Smitz et al. [42], and the two metrics were significantly and strongly correlated in this data set. The genetic differentiation between AENP and GNP/MNP ($D_{JOST}$ = 0.289/0.303, $F_{ST}$ = 0.255/0.257) was, to the best of our knowledge, the highest recorded (in terms of $F_{ST}$) between any two buffalo populations in southern Africa. It was substantially higher than the differentiation found between AENP and KNP ($F_{ST}$ = 0.131) by O'Ryan et al. [17]. It was also higher than the differentiation found between Hluhluwe-iMfolozi Park (HiP) and KNP ($D_{JOST}$ = 0.213, $F_{ST}$ = 0.141) by Smitz et al. [42] and between AENP and HiP ($F_{ST}$ = 0.227) estimated by O'Ryan et al. [17].

These levels of genetic differentiation between buffalo populations in South African protected areas are two to four times higher than the differentiation observed between Cape buffalo populations across Africa. AENP and HiP are more differentiated from KNP, than KNP is from populations in the East African countries of Kenya and Tanzania ($F_{ST}$ = 0.078–0.100) [42, 43]. Incredibly, southern African Cape buffalo (*S. c. caffer*) populations are more differentiated from each other than KNP buffalo are from forest buffalo (*S. c. nanus*) ($F_{ST}$ = 0.116–0.155), a different subspecies [43]. This amount of intraspecific genetic differentiation within the single subspecies of Cape buffalo populations in southern Africa (and particularly South Africa) is indicative of the highly fragmented nature of South Africa's protected areas and is another example of the human-induced population fragmentation and consequent reproductive isolation experienced by more than 25% of species in the world [47]. Thus, if past subdivision of South African buffalo populations was minimal, which is likely, as suggested by O'Ryan et al. [17], then action should be taken to re-establish gene flow [48], either through the establishment of natural corridors between populations or through translocations (e.g. from the disease-free GNP-MNP to AENP). Potential concerns about outbreeding depression and/or genomic swamping of the AENP buffalo population following this proposed genetic supplementation could be alleviated by considering the current support for an estimated short time since isolation (~100 years) [17, 49] and by conducting regular genetic monitoring of the population. Furthermore, recent studies showed local genes, and alleles related to local adaptation, were maintained in the recipient population after genetic supplementation, while the benefits of genetic rescue were still observed [50–52].

The results obtained with the STRUCTURE analyses showed that the removal of close relatives assisted in the more accurate identification of the most likely number of genetic clusters present in the data set, by satisfying the assumptions of the models implemented. The results suggested that these approaches were effective in mitigating the biases introduced into STRUCTURE analyses by the sampling of close relatives [53].

The individual assignment plots at $K$ = 3 illustrated the significant genetic differentiation between AENP and GNP-MNP. Apparent contributions to each other between these two clusters are most likely a result of shared ancestral polymorphism and not recent gene flow, given the small proportion of the contributions. At $K$ = 3, a contribution from a third gene pool (dark green) was prevalent in many individual buffalo in P001, P002, P004 and P007, and in a few individuals in the WPP population. The DAPC analysis also clustered a large proportion of individuals into this third gene pool. This third gene pool may have originated from one of two broadly defined regions: East Africa or the northern parts of southern Africa. We define East Africa here as the area north-north-east of Tanzania, inclusive. Thus, Tanzania, Kenya,

Burundi, Rwanda, Uganda and the southern parts of South Sudan, Ethiopia and Somalia. The support for an East African origin is that WPP had four buffalo potentially of Tanzanian or Kenyan origin introduced into its population via a Czech Republic zoo. Additionally, numerous private ranches in South Africa breed with East African buffalo, although the definition of East Africa may differ from ranch to ranch.

The northern parts of southern Africa are here defined as the "northern cluster" identified by Smitz et al. [42] and consists of northern Botswana, northern Zimbabwe, Angola and central and northern Mozambique (excluding Gorongosa National Park). Smitz et al. [42] identified two additional clusters; central (comprising southern Zimbabwe, southern Mozambique and northern South Africa, which included KNP- the source of GNP-MNP buffalo) and a southern cluster consisting mainly of buffalo from the isolated Hluhluwe-iMfolozi Park in the east of South Africa. The authors found evidence of the northern cluster gene pool in KNP, indicating gene flow between the northern cluster and KNP. It is thus conceivable that the signal of the third gene pool identified in the present study originated from northern Botswana/Zimbabwe buffalo and was also present in some of the buffalo that formed part of the disease-free breeding programme of KNP and therefore is present in both GNP-MNP and the private ranches (which source their buffalo from GNP-MNP, AENP and elsewhere). Some private ranches may also have independently introduced buffalo from this northern cluster to their properties, thus giving a more extensive signature of this third gene pool in the private ranch populations.

Both the above-mentioned scenarios are likely and may be occurring at the same time given the extensive translocations of buffalo in southern Africa. A more comprehensive data set consisting of Cape buffalo samples from across the subspecies' range, including private ranches, may reveal a more detailed genetic history regarding the origin of buffalo on private ranches and the natural genetic structure that exists across the continent.

In conclusion, we echo O'Ryan et al. [17] in recommending that the AENP population be augmented with buffalo of KNP origin to increase its genetic diversity- one breeding bull per generation (every ~7.5 years), to prevent genomic swamping and maintain unique diversity and local adaptation (if present) in AENP. This is more feasible now than in 1998, given that disease-free populations of KNP buffalo now exist in GNP and MNP and have high genetic diversity. Likewise, the population in WPP, with moderate genetic diversity and most of its founders from AENP, would benefit from the introduction of disease-free buffalo from GNP-MNP (at a similar frequency and rate to AENP) to firstly increase and then maintain genetic diversity. Disease-free buffalo populations (including private ranches, AENP, GNP-MNP and WPP) represent an important insurance policy for the species in southern Africa. Confirming that GNP, MNP and the private ranch populations maintained high genetic diversity was thus an important result. However, it is equally important to continually monitor genetic diversity of these populations, as the impact on genetic diversity of their recent establishment (i.e. founder effect) and fragmentation has likely not yet manifested. Furthermore, we found that genetic diversity, relatedness and inbreeding levels did not appear to be affected at the present time by the breeding strategies employed for buffalo on private ranches in this study, except perhaps for P006. It should be cautioned, however, that the genetic diversity on private ranches is highly dependent on the management practices on each ranch and the exchange of buffalo between ranches. If not managed actively and adequately, genetic diversity could be lost due to breeding practices, small population sizes (genetic drift) and inbreeding. We also showed that private ranch buffalo were predominantly of AENP and GNP-MNP origin, but that there was substantial contribution from a third, unsampled gene pool, most likely representing buffalo from East Africa, or the northern parts of southern Africa.

Private ranches contain a significant proportion of individuals of different wildlife species, not only in South Africa, but also in other countries such as Namibia, the USA and Spain [2,

5]. The present study is the first in southern Africa to evaluate the genetic diversity, at a large scale, of one of these species, the Cape buffalo, on private ranches compared to their source populations in national parks. Given that genetic diversity is a key component of biodiversity [54] and private ranches harbour a significant proportion of certain wildlife species, there is a clear case for similar studies to be conducted for other popular wildlife ranching species. This will require cooperation between private ranch owners, researchers and conservation authorities. The goal should be to catalogue the genetic diversity contained in private wildlife populations, as compared to national parks, for the benefit of the species. Furthermore, there appears to be no unified guidelines for private ranchers to assist in the genetic management of wildlife on their ranches in southern Africa to ensure sustainable use and long-term survival of the species [5]. Additional studies and appropriate guidelines are particularly pertinent given the increased pressures on extensive areas for ranching and conservation and on wildlife populations due to human expansion, anthropogenic activities and climate change [55, 56].

Future research on African buffalo could include investigating at a finer scale how specific ranching approaches (e.g. intensive vs. extensive, or particular breeding protocols) affect the genetic diversity, relatedness and inbreeding of populations and individual herds. Additionally, researchers could take advantage of the relative affordability of generating genome-wide data to investigate which loci might be under selection in private ranches and natural populations, to explore the demographic history of the species to investigate how it may have responded to historical climate change, to estimate divergence times of different populations, and to inspect what signatures historical population declines may have left in the genomes of individuals from different populations.

## Supporting information

**S1 Appendix. Supplementary methods.**
(PDF)

**S1 Fig. Estimated effective population size ($N_e$) of the buffalo population from each locality.** Vertical lines indicate 95% confidence intervals. Numbers inserted for GNP and GNP-MNP indicate the value of the upper bound of the 95% CI. The dashed line indicates the lower 95% CI of GNP. Values are also shown in S3 Table. PVT: Private ranches combined.
(PDF)

**S2 Fig. Statistical support for *K*.** The first column of graphs [L(*K*)] show the mean log likelihood of each value of *K* with its associated standard deviation, while the second column (Delta*K*) shows the most likely value of *K* as determined by the Evanno method. Rows indicate the full data set (FDS) and the relatives removed (RR) data set. The graphs were generated using StructureHarvester and further organized in Inkscape v0.92 (https://inkscape.org/).
(PDF)

**S3 Fig. Individual assignment plots of the STRUCTURE analyses at *K* = 2 and *K* = 3. A**–full data set, **B**–relatives removed. The plots were generated using the online version of Clumpak and further organized in Inkscape v0.92 (https://inkscape.org/).
(PDF)

**S4 Fig. Discriminant analysis of principal components (DAPC) of the full data set at *K* = 3.** AENP Cluster: Addo Elephant National Park cluster, GNP-MNP Cluster: Graspan and Mokala National Park cluster, "Other" Cluster: Third, unknown origin cluster.
(PDF)

**S1 Table. Summary statistics of microsatellite loci used in this study.** Calculated in Cervus v3.0.7.
(DOCX)

**S2 Table. Mean and variance of the relatedness estimators available in COANCESTRY.** TrioML (values in bold) had the lowest variance for each sampling locality and produces positive relatedness estimates between zero and one (as does DyadML).
(DOCX)

**S3 Table. Population summary statistics for each sampling locality.**
(DOCX)

**S4 Table. Relatedness and individual inbreeding statistics.**
(DOCX)

**S5 Table. Mean relatedness within sexes.**
(DOCX)

**S6 Table. Pairwise $D_{\text{JOST}}$ and $F_{\text{ST}}$ values with 95% confidence intervals.**
(DOCX)

**S7 Table. Hardy-Weinberg Equilibrium (HWE) probability tests of each sampling locality, with the full data set and relatives removed.** Data sets from sampling localities conformed to HWE after relatives were removed.
(DOCX)

**S8 Table. Original and Bonferroni-corrected linkage disequilibrium $p$-values of all pairs of loci in all sampling localities.** Both the full data set (FDS) and relatives removed (RR) data set are shown.
(XLSX)

## Acknowledgments

We acknowledge the contribution of Amy Clarke, previously of the Veterinary Genetics Laboratory in genotyping many of the samples over numerous years. We would like to thank Dave Zimmerman of SANParks for providing detailed population histories of the AENP, GNP and MNP buffalo populations. We also thank Mark Jago (previously of the Namibia Ministry of Environment and Tourism (MET)) and Malan Lindeque (Namibia MET), for providing a detailed population history of the WPP buffalo population and the MET for releasing this information for use in this research, as well as for providing samples. We thank the owners of the private ranches included in this study for their permission to publish the data in anonymized form.

## Author Contributions

**Conceptualization:** Deon de Jager, Cindy Kim Harper, Paulette Bloomer.

**Data curation:** Deon de Jager, Cindy Kim Harper.

**Formal analysis:** Deon de Jager.

**Funding acquisition:** Deon de Jager, Cindy Kim Harper, Paulette Bloomer.

**Investigation:** Deon de Jager.

**Methodology:** Deon de Jager.

**Project administration:** Deon de Jager, Cindy Kim Harper, Paulette Bloomer.

**Resources:** Cindy Kim Harper, Paulette Bloomer.

**Supervision:** Cindy Kim Harper, Paulette Bloomer.

**Validation:** Cindy Kim Harper, Paulette Bloomer.

**Visualization:** Deon de Jager.

**Writing – original draft:** Deon de Jager.

**Writing – review & editing:** Deon de Jager, Cindy Kim Harper, Paulette Bloomer.

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
