## [Decision Letter · Decision Letter 0]

26 Feb 2020

PONE-D-20-00888

Evolutionary and management implications of ranching and population fragmentation of Cape buffalo in southern Africa

PLOS ONE

Dear de Jager,

Thank you for submitting your manuscript to PLOS ONE. After careful consideration, we feel that it has merit but does not fully meet PLOS ONE’s publication criteria as it currently stands. Therefore, we invite you to submit a revised version of the manuscript that addresses the points raised during the review process.

The reviewers and I agree that the manuscript contains useful information, but it would benefit from a re-framing the focus of the manuscript, as suggested by Reviewer 2.

We would appreciate receiving your revised manuscript by Apr 11 2020 11:59PM. To enhance the reproducibility of your results, we recommend that if applicable you deposit your laboratory protocols in protocols.io, where a protocol can be assigned its own identifier (DOI) such that it can be cited independently in the future. For instructions see: http://journals.plos.org/plosone/s/submission-guidelines#loc-laboratory-protocols

We look forward to receiving your revised manuscript.

Kind regards,

Elissa Z. Cameron

Academic Editor

PLOS ONE

Journal Requirements:

2. In your Methods section, please provide additional location information of the study sites, including geographic coordinates for the data set if available.

Reviewers' comments:

Reviewer's Responses to Questions

**Comments to the Author**

1. Is the manuscript technically sound, and do the data support the conclusions?

Reviewer #1: Yes

Reviewer #2: Partly

2. Has the statistical analysis been performed appropriately and rigorously? 

Reviewer #1: Yes

Reviewer #2: Yes

3. Have the authors made all data underlying the findings in their manuscript fully available?

Reviewer #1: Yes

Reviewer #2: Yes

4. Is the manuscript presented in an intelligible fashion and written in standard English?

Reviewer #1: Yes

Reviewer #2: Yes

5. Review Comments to the Author

Reviewer #1: This is a well-written manuscript which describes genetic variation in buffalo in private ranches in SA, as well as Graspan and Mokala NP populations which which were stocked through a breeding programme of Kruger NP buffalo, to establish disease-free populations with high genetic diversity. Populations on 12 private ranches in South Africa and Waterberg Plateau NP (Namibia) were measured, which established predominantly with founders from Addo, Graspan and Mokala, with some supplementation from zoos.

The question in my mind relates not to the quality of the work, which is sound, but to how broadly relevant it would be to readers of PLOSone. It might be better-suited to a journal such as the SA Journal of Wildlife Research.

Reviewer #2: This is a review of the manuscript PONE-D-20-00888 by de Jager et al. The manuscript presents a very large data set of microsatellite genotypes from South African buffalo populations representing national parks, semi-managed populations and private game ranches. The authors conduct a thorough set of analyses to infer whether genetic diversity is comparable in different classes of populations and to elucidate the genetic structure among populations.

While the analyses are generally competently carried out and an impressive number of complementary approaches are taken to verify various results, I think the manuscript suffers from trying to wring too much out of the data. The problem is that many of the analyses are based on assumptions that are blatantly violated in the special case of intensively managed or newly founded populations. It is also as a consequence too long, since the results can be boiled down to a very few bullet points, some of which are not new observations. Finally, there are problems and concerns with some of the analyses. I recommend the authors to reshape the manuscript focusing exclusively on 1) genetic diversity in the different populations, 2) inbreeding and 3) relatedness. These are the types of statistics that are informative and relevant for comparing the private and other populations. I therefore recommend that the authors submit a significantly abbreviated and modified version of the manuscript for review. My recommendations are provided in detail below.

1) The Abstract is too long (although just within the journal limits) and contains several items of “Introduction” nature, such as detailed background about the provenance of each population which cannot be readily assimilated by the reader at this early point.

2) The analyses presented in the section Population structure from L192 onwards are in my opinion not relevant, except perhaps the Fst calculations. The Structure analyses are problematic because the basic assumptions of the method are clearly violated. Structure assumes that K ancestral populations exist, from which the samples have their ancestry and which behaves in other respects like a WF population. This is clearly not the case in the complex network of the sampled populations, in which reticulated gene flow, founder events and in some cases selective breeding takes place. The authors go through considerable and commendable efforts in these analyses, but in my opinion it is very doubtful whether the results are meaningful. This is also reflected in the difficulty in establishing the “correct” K. At the very least the Structure plots are impossible to interpret. For example, why would both AENP and GNP/MNP contain admixture from each other? Why (if the authors’ speculation that the green cluster at K3 corresponds to an East African contribution) would GNP and MNP also carry this signal, albeit to a smaller extent? The authors should in my opinion either i) remove these analyses, ii) move them to a Supplementary document and significantly tone down the conclusions based on them or iii) convince me that these analyses tell us anything useful.

3) The estimation of the effective population size also does not convince me. Although I do not know the details of the LD based method, I doubt that the estimated Ne’s tell us anything useful about these populations. They all range within very similar values. The actual observed diversity statistics Ar and het are in my opinion more informative, again because it is doubtful whether these populations behave in any way in accordance with the underlying assumption of any population genetic model.

4) As above, the bottleneck estimation analysis is not very relevant. Due to the complex history and patchy ancestry of most populations (at least the private ranch populations), it is highly dubious whether we could reasonably expect a population genetically determined equilibrium correspondence between Ar and het. I would say that in all populations but AENP, this test is not meaningful.

5) I would have liked some further discussion about the breeding practice on the private ranches, and how this relates to the observed statistics. For example I would expect very high relatedness in the private ranches, given that all or most individuals probably share their father. Kinship coefficients are not discussed in the paper, although they were estimated as part of the downsampling procedure in the Structure analysis. I would say that a different relatedness pattern in farms relative to wild populations is one of the most obvious differences to look for, and the study should have this as one of the main aims. In fact, kinship coefficient distribution could be highly informative regarding the breeding practice in private and wild populations. For the same reason I think it quite surprising that private farms do not show any (substantial) difference in either Ne, het or Ar. The only reason I can think of is that wild buffalo populations approximate a similar mating system as that on the private ranches, but I don’t think this is supported by data on wild buffalo social structure.

6) Why did the authors not include some samples from eastern African populations, if they specifically want to assess whether there is East African ancestry in some populations? Such samples exist and are relatively abundant. There are even published microsat data sets, presumably (although I did not check) using the same microsatellite markers.

7) I did not see the authors declare any conflict of interest. It needs to be evaluated how the mixing of commercial genotyping paid for by the private ranches has influenced any study decision.

6. PLOS authors have the option to publish the peer review history of their article (what does this mean?). If published, this will include your full peer review and any attached files.

Reviewer #1: No

Reviewer #2: No

---

## [Author Response · Author response to Decision Letter 0]

18 Apr 2020

Response to academic editor and reviewers

We would like to thank the two reviewers for investing their time in reviewing our manuscript. We appreciate the constructive and detailed feedback provided by Reviewer 2 in particular. We also thank the academic editor, Professor Elissa Z. Cameron, for her time and efforts in handling this manuscript.

Editor comments:

Journal Requirements:

>>>Response:

• Noted. Apologies for previous deviations from the style and/or file naming requirements.

2. In your Methods section, please provide additional location information of the study sites, including geographic coordinates for the data set if available.

>>>Response:

• Additional location information has been included, together with GPS coordinates of the National Parks. Private ranch locations were omitted as this may compromise the anonymity of the participating ranches. The location of the ranches was not important for the analyses and conclusions of the study, except to say that all ranches are located in South Africa. A statement to this effect has been included in the Methods section.

• It is unclear whether the geographic coordinates of individual samples from the National Parks were recorded upon collection and if so, these were not made available to us by the conservation authorities who performed the sampling. However, we felt that such fine scale GPS data were not needed to answer the questions of this study.

Reviewer 1:

Reviewer #1: This is a well-written manuscript which describes genetic variation in buffalo in private ranches in SA, as well as Graspan and Mokala NP populations which which were stocked through a breeding programme of Kruger NP buffalo, to establish disease-free populations with high genetic diversity. Populations on 12 private ranches in South Africa and Waterberg Plateau NP (Namibia) were measured, which established predominantly with founders from Addo, Graspan and Mokala, with some supplementation from zoos.

The question in my mind relates not to the quality of the work, which is sound, but to how broadly relevant it would be to readers of PLOSone. It might be better-suited to a journal such as the SA Journal of Wildlife Research.

>>>Response:

• We appreciate the reviewer’s comments on the manuscript. We refer to the stated scope of PLOS ONE (https://journals.plos.org/plosone/s/journal-information#loc-scope), which we believe our manuscript satisfies: 

“PLOS ONE welcomes original research submissions from the natural sciences, medical research, engineering, as well as the related social sciences and humanities, including:

o Primary research that contributes to the base of scientific knowledge, including interdisciplinary, replication studies, and negative or null results.”

And (https://journals.plos.org/plosone/static/publish)”

“We evaluate research on scientific validity, strong methodology, and high ethical standards—not perceived significance.”

Reviewer 2:

Reviewer #2: This is a review of the manuscript PONE-D-20-00888 by de Jager et al. The manuscript presents a very large data set of microsatellite genotypes from South African buffalo populations representing national parks, semi-managed populations and private game ranches. The authors conduct a thorough set of analyses to infer whether genetic diversity is comparable in different classes of populations and to elucidate the genetic structure among populations.

While the analyses are generally competently carried out and an impressive number of complementary approaches are taken to verify various results, I think the manuscript suffers from trying to wring too much out of the data. The problem is that many of the analyses are based on assumptions that are blatantly violated in the special case of intensively managed or newly founded populations. It is also as a consequence too long, since the results can be boiled down to a very few bullet points, some of which are not new observations. Finally, there are problems and concerns with some of the analyses. I recommend the authors to reshape the manuscript focusing exclusively on 1) genetic diversity in the different populations, 2) inbreeding and 3) relatedness. These are the types of statistics that are informative and relevant for comparing the private and other populations. I therefore recommend that the authors submit a significantly abbreviated and modified version of the manuscript for review. My recommendations are provided in detail below.

>>>Response:

• We thank the reviewer for these suggestions to focus the content and message of the manuscript. We have taken these comments on board and focused the manuscript on four analyses: 1) Genetic diversity, 2) Relatedness, 3) Inbreeding and 4) Population differentiation. We provide more detailed responses to the reviewer’s comments below.

1) The Abstract is too long (although just within the journal limits) and contains several items of “Introduction” nature, such as detailed background about the provenance of each population which cannot be readily assimilated by the reader at this early point.

>>>Response:

• The abstract has been shortened by removing the background of the provenance of each population. New results about relatedness and inbreeding were also incorporated into the abstract.

2) The analyses presented in the section Population structure from L192 onwards are in my opinion not relevant, except perhaps the Fst calculations. The Structure analyses are problematic because the basic assumptions of the method are clearly violated. Structure assumes that K ancestral populations exist, from which the samples have their ancestry and which behaves in other respects like a WF population. This is clearly not the case in the complex network of the sampled populations, in which reticulated gene flow, founder events and in some cases selective breeding takes place. The authors go through considerable and commendable efforts in these analyses, but in my opinion it is very doubtful whether the results are meaningful. This is also reflected in the difficulty in establishing the “correct” K. At the very least the Structure plots are impossible to interpret. For example, why would both AENP and GNP/MNP contain admixture from each other? Why (if the authors’ speculation that the green cluster at K3 corresponds to an East African contribution) would GNP and MNP also carry this signal, albeit to a smaller extent? The authors should in my opinion either i) remove these analyses, ii) move them to a Supplementary document and significantly tone down the conclusions based on them or iii) convince me that these analyses tell us anything useful.

>>>Response:

• We do not entirely agree with the reviewer’s comments on the population structure analyses. While we agree that the Fst calculations are relevant, particularly for comparative purposes to previous (and future) studies, we think the Structure analyses also contribute some useful information/results. At the very least, it provides genetic support (for the first time) for the reported origin of buffalo on private ranches in South Africa, namely Addo and Kruger at least, and yes, a third gene pool unsampled in this study. 

• We heeded the reviewer’s comments about the difficulty in establishing the “correct” K and removed all the Structure analyses except for the full data set and the relatives removed data set. While we wanted to show thoroughness in our exploration of the data and genetic structure, we lament that this came across as a difficulty in establishing the “correct” K, as this was not the intention. Thus, we focused the analysis on just the removal of relatives and the full data set. We justify our use of the “relatives removed” data set by showing that in this data set all locus pairs are in linkage equilibrium and all populations are in Hardy-Weinberg equilibrium (S7 Table and S8 Table), which are the assumptions made about the underlying data by Structure [1]. We concede that the full data set did not entirely conform to these assumptions. However, we are confident that K = 3 is the most accurate K we can estimate with this data set.

• Consequently, we have retained this analysis in the manuscript, but have moved the majority of the methods describing this analysis to S1 Appendix and moved the results to the supplementary information (S2 Fig, S3 Fig, S4 Fig – the DAPC analysis). We have cut down the discussion of the structure results substantially (L467-490). Furthermore, we provide explanations for the features of the assignment plots highlighted by the reviewer. We suggest what appears to be low levels of admixture between AENP and GNP-MNP is most likely shared ancestral polymorphism (L473) and not recent gene flow, particularly since this signal is very low at K = 3. Furthermore, we provide an explanation as to why GNP-MNP would contain a signal from the third cluster (perhaps East Africa), but also tone down the conviction with which we state this to reflect the possibility that this signal might be of proper East African origin, but is at least of non-South African origin (L483-490). We think this additional support strengthens our argument that private ranches consist of three gene pools and does not weaken this argument.

• We hope that the adjustments we made to the structure analyses and conclusions convince the reviewer of their robustness and provides enough support to include these analyses in the manuscript.

3) The estimation of the effective population size also does not convince me. Although I do not know the details of the LD based method, I doubt that the estimated Ne’s tell us anything useful about these populations. They all range within very similar values. The actual observed diversity statistics Ar and het are in my opinion more informative, again because it is doubtful whether these populations behave in any way in accordance with the underlying assumption of any population genetic model.

>>>Response:

• We note this concern and have significantly reduced the presentation of these results (moved to supplementary information, S1 Fig) and the discussion of Ne. We also included a statement (L406) reflecting the reviewer’s concern about the violation of population genetics models of the private ranches in particular, and we agree with the reviewer in this respect. We have, however, retained the discussion of Ne for the national parks, as we think it is a valuable analysis for these unmanaged, or semi-managed populations, particularly for comparisons to previous (and future) studies.

4) As above, the bottleneck estimation analysis is not very relevant. Due to the complex history and patchy ancestry of most populations (at least the private ranch populations), it is highly dubious whether we could reasonably expect a population genetically determined equilibrium correspondence between Ar and het. I would say that in all populations but AENP, this test is not meaningful.

>>>Response:

• We agree with the reviewer on this point and have removed the bottleneck estimation analysis and corresponding discussion.

5) I would have liked some further discussion about the breeding practice on the private ranches, and how this relates to the observed statistics. For example I would expect very high relatedness in the private ranches, given that all or most individuals probably share their father. Kinship coefficients are not discussed in the paper, although they were estimated as part of the downsampling procedure in the Structure analysis. I would say that a different relatedness pattern in farms relative to wild populations is one of the most obvious differences to look for, and the study should have this as one of the main aims. In fact, kinship coefficient distribution could be highly informative regarding the breeding practice in private and wild populations. For the same reason I think it quite surprising that private farms do not show any (substantial) difference in either Ne, het or Ar. The only reason I can think of is that wild buffalo populations approximate a similar mating system as that on the private ranches, but I don’t think this is supported by data on wild buffalo social structure.

>>>Response:

• We appreciate this comment, as we think the analyses that we conducted to address it has added a lot of value to the manuscript. We have included substantial analyses of relatedness and inbreeding (Methods: L181-204, Results: L251-293, incl. Figs 2, 3 and 4). The discussions of these results are scattered throughout the Discussion, but the main part is from L410-439. The discussions of these results had to be kept at high level, as information of the precise breeding protocols on each private ranch was not available for this study. We did, however, include a fine-scale study such as that as a future perspective.

• While one might expect a statistical significance test of a difference in mean relatedness between localities, the L-shaped distributions of pairwise relatedness values meant that the usual statistical tests for comparing distributions were not necessarily applicable here [2]. Instead, we compared the shape and tails of the distributions, which we think were more informative about the distributions of relatedness than just the mean (although we do report the mean as well). 

• Similarly, with the distributions of individual inbreeding estimates, we think that the actual value of the mean is what was important, as the magnitude of this value has direct biological consequences, whereas measures of statistical differences between the means were not necessarily as informative. Particularly because most populations had low inbreeding.

6) Why did the authors not include some samples from eastern African populations, if they specifically want to assess whether there is East African ancestry in some populations? Such samples exist and are relatively abundant. There are even published microsat data sets, presumably (although I did not check) using the same microsatellite markers.

>>>Response:

• Unfortunately, we did not have access to East African samples during this study (and at present also do not have access to such samples). 

• It was not an original aim of the study to assess whether there is East African ancestry in some populations, but it rather came about as an after-the-fact potential explanation, once we found K = 3 was the optimal K. However, we concede that we subsequently probably gave this result more importance than was required in the context of the whole study.

• We considered using published microsatellite data sets where a few loci do overlap with those in this study. However, we decided against this approach, because one of the major pitfalls of microsatellites as genetic markers is the difficulty to combine data sets of separate studies, or even the same samples genotyped in different labs. This is because there are many factors that can influence the scoring of alleles, such as the sequencing machine used, the polymer used within the capillaries of the machine, the reading/scoring rules such as binning, allele naming, how stutter peaks are dealt with and human error in scoring alleles. Therefore, it is difficult to reconcile microsatellite genotype data sets of different studies on the same species or even populations, unless the raw data is available for re-scoring. We did not want to introduce this uncertainty into the current study and so decided against using published data sets of East African buffalo here. However, perhaps an opportunity exists for a separate study to properly evaluate whether combining buffalo microsatellite data sets is feasible in this case.

7) I did not see the authors declare any conflict of interest. It needs to be evaluated how the mixing of commercial genotyping paid for by the private ranches has influenced any study decision.

>>>Response:

• We acknowledge this concern and we can honestly state that the private ranches had no input into the study design or conclusions, and all agreed to participate in the study regardless of the outcome, provided their identities remained anonymous. Therefore, we declared no conflict of interest. However, we can include the above statement in the conflict of interest section.

References

1. Pritchard JK, Stephens M, Donnelly P. Inference of population structure using multilocus genotype data. Genetics. 2000;155(2):945-59. PubMed PMID: PMC1461096.

2. Bradley JV. The insidious L-shaped distribution. Bulletin of the Psychonomic Society. 1982;20(2):85-8. doi: 10.3758/BF03330089.

---

## [Decision Letter · Decision Letter 1]

20 May 2020

PONE-D-20-00888R1

Genetic diversity, relatedness and inbreeding of ranched and fragmented Cape buffalo populations in southern Africa

PLOS ONE

Dear Dr de Jager,

Thank you for submitting your manuscript to PLOS ONE. After careful consideration, we feel that it has merit but does not fully meet PLOS ONE’s publication criteria as it currently stands. Therefore, we invite you to submit a revised version of the manuscript that addresses the points raised during the review process.

Thank you for taking the time to thoroughly revise the manuscript, which has been substantially improved. The reviewers raises some further minor points that will need to be addressed.

We would appreciate receiving your revised manuscript by Jul 04 2020 11:59PM. To enhance the reproducibility of your results, we recommend that if applicable you deposit your laboratory protocols in protocols.io, where a protocol can be assigned its own identifier (DOI) such that it can be cited independently in the future. For instructions see: http://journals.plos.org/plosone/s/submission-guidelines#loc-laboratory-protocols

We look forward to receiving your revised manuscript.

Kind regards,

Elissa Z. Cameron

Academic Editor

PLOS ONE

Reviewers' comments:

Reviewer's Responses to Questions

**Comments to the Author**

1. If the authors have adequately addressed your comments raised in a previous round of review and you feel that this manuscript is now acceptable for publication, you may indicate that here to bypass the “Comments to the Author” section, enter your conflict of interest statement in the “Confidential to Editor” section, and submit your "Accept" recommendation.

Reviewer #3: All comments have been addressed

2. Is the manuscript technically sound, and do the data support the conclusions?

Reviewer #3: Yes

3. Has the statistical analysis been performed appropriately and rigorously? 

Reviewer #3: Yes

4. Have the authors made all data underlying the findings in their manuscript fully available?

Reviewer #3: No

5. Is the manuscript presented in an intelligible fashion and written in standard English?

Reviewer #3: Yes

6. Review Comments to the Author

Reviewer #3: The manuscript is very clear in its objectives and the authors properly investigated their question using appropriate statistical tests. Even if I wasn’t personally involved in the first review process, I can see that it was already greatly improved. Results are also properly presented. I feel like the research study is worth being published in PloS One, both considering the investigated number of specimens/ DNA markers and the research question. However, I have some comments which I would like to be addressed first.

Material and methods:

-large banks of tissue and blood from wildlife do exist in southern Africa, collected and preserved since decades. The authors are not specifying anything about the collection timing of the involved samples. Were the specimens sampled in a fair limited time period? And is this time period recent and comparable between populations? Further information regarding this point are essential to check if the authors properly interpreted their results. This could greatly impact the discussion.

-why I fully understand to respect the private game ranches anonymity, I would still like that the authors present some extra numbers. Especially, the current Nc and the year when the ranches acquired their first buffalo on their property are in my opinion important for the discussion (considering the fairly long generation time (5-7 years) of buffaloes).

-L144: you refer to “KNP breeding group”. What does this refer to? Wild herds or do they have kind of isolated groups of individuals in the NP from where they select the individuals which will be translocated?

Results:

-L220: “seventh largest sample size”: is this referring to all south African buffalo populations or the one sampled in your study?

-L311: so about 10% were removed, can you provide some numbers?

-Bonferroni corrections: standard or sequential?

Discussion

-both considering the large number of private alleles and the STRUCTURE results, I understand the proposed hypothesis: an origin from an unsampled population. Why, however, focus on East Africa? What do you actually understand by East Africa? Something that wasn’t discussed is the fact that in southern African buffalo populations, three population clusters based on microsatellite genotypes were identified in Smitz et al. 2014. Considering that translocation is common practice in southern Africa, both between NP and ranches (and between them), could your signal not originate from the unsampled southern cluster (North Zimbabwe, Botswana)? Especially, a lot of buffalo from Hwange were translocated since decades, as this was also a disease free buffalo stock...

-statements L340 and L489: ok but important to specify that a re-evaluation will be needed when required. My point is that buffaloes have retained a great part of their historical genetic diversity – still recorded now, even after the severe Nc loss linked to the diseases outbreaks (multiple studies stating this). However, impact of the fragmentation of its habitat – which is quite “recent”, is not yet genetically measurable. Probably because not enough generations have elapsed since fragmentation has started.

Also, its a vice versa situation. For now, if there is no external genetic inputs on private ranches, you also expect a decrease of the genetic diversity in the future. So it will highly depend on the management practices. Are some general long-term guidelines or regulations existing for the management of wildlife in private ranches in southern Africa?

Conclusion: To make such statements, you should consider the time in generations which elapsed since the private ranching practices have started to develop in southern Africa (info needed per ranches in the M&M). I would also suggest to be much more cautious when stating that genetic diversity etc. were not affected by the breeding strategies employed (L492). Not yet!

-L373: careful with this statement, increased genetic diversity is not always beneficial: outbreeding, loss of local adaptations ... Why for example does the Eastern African buffalo males have longer horns? I guess it is not plasticity or there would be no point in translocating individuals from East to South.

-L378: see the importance to provide the Nc of the private ranches. Number of sampled specimens is not equivalent.

-relatedness discussion: has to be re-evaluated in regard of the collection timing. Also, consider discussing the impact of the migration of only a few individuals per generation on the genetic structure of a population- literature about this aspect exist.

-L445: an important point is the use of FST as differentiation indices. This is tricky, especially when comparing different subspecies. FST is not a measure of “similarity”, it’s based on allelic frequencies. For example: if two populations are fixed for alternative alleles, then pairwise FST = 1. But if you have two populations, one with 50% aa and 50% bb, the other with 50% cc and 50% dd, then their pairwise FST = 0.33. Are the two last population genotypically closer to each other than the two first populations?

General minor:

-provide an access to your database – dryad or as supp material

-check number of decimals you are using across the manuscript, ex. Line 249, 444.

-“P” or “p” -value, homogenize

-line 295-296: numbers are not essential, the most important is that they are not significant.

-line 299: how did you calculated the average? Did you included the non significant estimates?

7. PLOS authors have the option to publish the peer review history of their article (what does this mean?). If published, this will include your full peer review and any attached files.

Reviewer #3: No

---

## [Author Response · Author response to Decision Letter 1]

11 Jul 2020

Response to reviewers

We would like to thank Reviewer 3 for their constructive comments on the manuscript. We believe the changes we implemented further improves the manuscript. Please see our responses below and note that all line numbers given in our responses refer to lines in the document with track changes (“Revised Manuscript with Track Changes.docx”).

Reviewer 3

The manuscript is very clear in its objectives and the authors properly investigated their question using appropriate statistical tests. Even if I wasn’t personally involved in the first review process, I can see that it was already greatly improved. Results are also properly presented. I feel like the research study is worth being published in PloS One, both considering the investigated number of specimens/ DNA markers and the research question. However, I have some comments which I would like to be addressed first.

Material and methods:

• large banks of tissue and blood from wildlife do exist in southern Africa, collected and preserved since decades. The authors are not specifying anything about the collection timing of the involved samples. Were the specimens sampled in a fair limited time period? And is this time period recent and comparable between populations? Further information regarding this point are essential to check if the authors properly interpreted their results. This could greatly impact the discussion.

Response:

>>> Sampling timing has been added to the manuscript (L107-109 and Table 1). All samples were collected within one buffalo generation (2008 – 2015) and 99% of the samples were collected between 2011 and 2015. We think this time period is recent enough, since only one to two buffalo generations have passed between sample collection and the writing of this manuscript. There is considerable overlap and/or few years difference regarding the timing of collection between populations (Table 1). Considering all the above, we are of the opinion that the collection timing does not have an impact on the discussion. We wrote the manuscript with this knowledge in mind but should have shared this information with the reader and thus thank the reviewer for pointing it out.

• why I fully understand to respect the private game ranches anonymity, I would still like that the authors present some extra numbers. Especially, the current Nc and the year when the ranches acquired their first buffalo on their property are in my opinion important for the discussion (considering the fairly long generation time (5-7 years) of buffaloes).

Response:

>>> We understand that the current Nc of private ranches would be interesting, particularly in the discussion of Ne. However, Reviewer 2 (in the previous round) expressed reservations about the Ne analysis and its discussion for the private ranches, which we subsequently toned down and shortened significantly, with a warning to interpret with caution. We do not think that Nc information would change this, as it does not change the nature of private ranch populations, which was the basis of Reviewer 2’s reservations. Furthermore, we unfortunately do not have the census data at hand for each private ranch and it may take months, or longer, to obtain this information from the owners, if it is recorded. Thus, on balance, we do not think that adding the census information would add substantial value to this study.

The date at which buffalo were first acquired by the various ranches may assist in interpreting the observed genetic diversity statistics, but considering the dynamic nature of buffalo populations on private ranches (buying and selling of buffalo), the year of establishment alone would, in our opinion, not be sufficient to properly interpret the genetic diversity in the manner we think the reviewer is suggesting. The lack of knowledge about the individual history of each private ranch is why we kept our discussion and analyses at a fairly high level, as we could not dive into the details of each ranch. We believe that it is beyond the scope of this study, but that it should rather be the focus of a separate study detailing more precisely how the individual histories of each private ranch (if this information can be obtained) affects the genetic diversity and relatedness of buffalo on each ranch.

• L144: you refer to “KNP breeding group”. What does this refer to? Wild herds or do they have kind of isolated groups of individuals in the NP from where they select the individuals which will be translocated?

Response:

>>> The breeding group consisted of ~140 buffalo cows and 10 bulls mainly from northern KNP (originally wild herds), that were contained in a fenced-off camp in the NP. This information has been added to the manuscript (L146 – 148). 

Results:

• L220: “seventh largest sample size”: is this referring to all south African buffalo populations or the one sampled in your study?

Response:

>>> In the study. An edit was made to reflect this (L228).

• L311: so about 10% were removed, can you provide some numbers?

Response:

>>> The sample sizes before and after removal of relatives are provided in S7 Table (referenced a couple of sentences later). A line was also added to S1 Appendix (supplementary methods) after describing how relatives were removed to direct the reader to S7 Table for sample sizes before and after removal of relatives. The proportion removed varies for each sampling locality, depending on the relatedness levels observed.

• Bonferroni corrections: standard or sequential?

Response:

>>> Standard Bonferroni corrections were performed. This clarification was added to S1 Appendix, where the HWE analysis is described.

Discussion

• both considering the large number of private alleles and the STRUCTURE results, I understand the proposed hypothesis: an origin from an unsampled population. Why, however, focus on East Africa? What do you actually understand by East Africa? Something that wasn’t discussed is the fact that in southern African buffalo populations, three population clusters based on microsatellite genotypes were identified in Smitz et al. 2014. Considering that translocation is common practice in southern Africa, both between NP and ranches (and between them), could your signal not originate from the unsampled southern cluster (North Zimbabwe, Botswana)? Especially, a lot of buffalo from Hwange were translocated since decades, as this was also a disease free buffalo stock...

Response:

>>> The reviewer makes a good point here. We now define our understanding of East Africa in our discussion. Furthermore, we explicitly include in our discussion the possibility that the third signal may be from the northern cluster (North Zimbabwe, Botswana; the southern cluster is Hluhluwe-iMfolozi Park) as defined by Smitz et al. 2014. We previously referred to this possibility using the term “non-South African origin” but we now explicitly refer to the clusters identified by Smitz et al. 2014 as suggested by the reviewer, which we think improves the discussion. (L507-535). We also adjust the population history of the private ranches in the Methods section to reflect this other potential source (L169, L172).

[The reason we focused on East Africa is because of the East African origin of a few buffalo introduced into WPP (most likely Kenya or Tanzania – this is mentioned in the population history section of the methods), which also has the signal from the third gene pool. Additionally, some private ranches have introduced East African buffalo to their properties and advertise the fact they are breeding with these buffalo, although, admittedly, we do not know what their definition of East Africa is. East Africa for us is defined by the southern border of Tanzania. Thus, countries north/north-east of Tanzania are included, but mainly Tanzania and Kenya. Private buffalo owners may each be working under a different definition of East Africa, which opens this term up for interpretation and complicates genetics conclusion].

• statements L340 and L489: ok but important to specify that a re-evaluation will be needed when required. My point is that buffaloes have retained a great part of their historical genetic diversity – still recorded now, even after the severe Nc loss linked to the diseases outbreaks (multiple studies stating this). However, impact of the fragmentation of its habitat – which is quite “recent”, is not yet genetically measurable. Probably because not enough generations have elapsed since fragmentation has started. Also, its a vice versa situation. For now, if there is no external genetic inputs on private ranches, you also expect a decrease of the genetic diversity in the future. So it will highly depend on the management practices. Are some general long-term guidelines or regulations existing for the management of wildlife in private ranches in southern Africa?

Response:

>>> Yes, we agree that it is important to monitor genetic diversity of these populations. We have included a statement to highlight this important point (L557-559). The private ranches comment is addressed in the response to the next comment. 

• Conclusion: To make such statements, you should consider the time in generations which elapsed since the private ranching practices have started to develop in southern Africa (info needed per ranches in the M&M). I would also suggest to be much more cautious when stating that genetic diversity etc. were not affected by the breeding strategies employed (L492). Not yet!

Response:

>>> We amended our conclusion related to the private ranches, thereby addressing this comment and the previous comment (L561-565). Since we do not have (and did not include) the date of establishment of buffalo populations on various private ranches, we made a more general cautionary statement on the importance of active management of buffalo on private ranches. Presently, there seems to be no unified regulations that exist for the management of wildlife on private ranches in southern Africa. We note this on L578-581.

• L373: careful with this statement, increased genetic diversity is not always beneficial: outbreeding, loss of local adaptations ... Why for example does the Eastern African buffalo males have longer horns? I guess it is not plasticity or there would be no point in translocating individuals from East to South.

Response:

>>> A qualifying statement has been added here (L386-388). Outbreeding depression/loss of local adaptation is also addressed on L490-495 in relation to genetic supplementation of AENP. 

[This comment relates to WPP, which is a population established by Namibian authorities with buffalo from AENP and other sources. Thus, local adaptation has likely not yet occurred here and the probability of outbreeding depression is low between populations within the Cape buffalo subspecies in general, if one considers the guidelines suggested by Frankham et al. (1)].

• L378: see the importance to provide the Nc of the private ranches. Number of sampled specimens is not equivalent.

Response:

>>> We added an extension of this explanation (L393-394). The point made in this line refers to the fact that the sample size from P007 is small (not that the population is small), thereby implying that this sample size may not be as statistically robust as the sample sizes from the other localities, which then led to the significantly negative FIS value. It is an explanation of a statistical nature rather than a biological one. Whether the samples adequately represent the population on this ranch, depending on the census size, is not what the statement was addressing.

• relatedness discussion: has to be re-evaluated in regard of the collection timing. Also, consider discussing the impact of the migration of only a few individuals per generation on the genetic structure of a population- literature about this aspect exist.

Response:

>>> As stated in response to the reviewer’s first comment, we are of the opinion that the collection timing does not have a significant influence, if any, on the discussion, given that all samples were collected in the space of one buffalo generation. The relatedness discussion was, however, restructured and amended to consider the effect of migrants on relatedness distributions- an explanation that we had overlooked. L429-444.

• L445: an important point is the use of FST as differentiation indices. This is tricky, especially when comparing different subspecies. FST is not a measure of “similarity”, it’s based on allelic frequencies. For example: if two populations are fixed for alternative alleles, then pairwise FST = 1. But if you have two populations, one with 50% aa and 50% bb, the other with 50% cc and 50% dd, then their pairwise FST = 0.33. Are the two last population genotypically closer to each other than the two first populations? 

Response:

>>> We agree with the reviewer on this point and we are aware of this fact about FST. We used FST for differentiation for compatibility with historical studies where DJOST (a true differentiation measure) was not calculated. We thus also report DJOST, because it actually estimates differentiation. We added to the manuscript a correlation test between FST and DJOST and show significant and strong correlation between the two measures (Spearman’s rho = 0.90, p-value < 2.2e-16). We think this adds some rigour to the use of FST as a proxy for differentiation for comparisons with previous studies, which we think is important to include in the discussion. This analysis is reported in the methods (L212-215), the results (L301-302) and the discussion (L467-469).

General minor:

• provide an access to your database – dryad or as supp material

Response:

>>> There should have been a link to the data set on Dryad. We thought we had provided it on the online submission system. We will check with the editor to ensure that the link is included in the final manuscript.

• check number of decimals you are using across the manuscript, ex. Line 249, 444.

Response:

>>> Thank you for pointing this out. We have checked and made sure we use the same number of decimals within each metric.

• “P” or “p” -value, homogenize

Response:

>>> Done.

• line 295-296: numbers are not essential, the most important is that they are not significant.

Response:

>>> Noted. Adjustments were made to these lines (L306-308).

• line 299: how did you calculated the average? Did you included the non significant estimates?

Response:

>>> All instances of “average” was replaced with the term “mean”, which is the most accurate term to use in this case, since it inherently provides information on how the “average” was calculated. Non-significant estimates were included, yes. The vast majority of pairwise estimates were significant.

References

1. Frankham R, Ballou JD, Eldridge MDB, Lacy RC, Ralls K, Dudash MR, et al. Predicting the probability of outbreeding depression. Conserv Biol. 2011;25(3):465-75. doi: 10.1111/j.1523-1739.2011.01662.x.

---

## [Editor Report · Decision Letter 2]

14 Jul 2020

Genetic diversity, relatedness and inbreeding of ranched and fragmented Cape buffalo populations in southern Africa

PONE-D-20-00888R2

Dear Dr. de Jager,

We’re pleased to inform you that your manuscript has been judged scientifically suitable for publication and will be formally accepted for publication once it meets all outstanding technical requirements.

Kind regards,

Elissa Z. Cameron

Academic Editor

PLOS ONE
---

## [Editor Report · Acceptance letter]

17 Jul 2020

PONE-D-20-00888R2 

Genetic diversity, relatedness and inbreeding of ranched and fragmented Cape buffalo populations in southern Africa 

Dear Dr. de Jager:

I'm pleased to inform you that your manuscript has been deemed suitable for publication in PLOS ONE. Congratulations! Your manuscript is now with our production department. 

Kind regards, 

on behalf of

Prof Elissa Z. Cameron 

Academic Editor

PLOS ONE